# Collecting genetic samples and linked mental health data from adolescents in schools: protocol coproduction and a mixed-methods pilot of feasibility and acceptability

Naomi Warne [iD],[1,2] Sarah Rook,[1] Rhys Bevan Jones,[1] Rachel Brown,[3] Lesley Bates,[1] Lucinda Hopkins-Jones,[1] Alexandra Evans,[1] Jeremy Hall,[1] Kate Langley,[1,4] Anita Thapar,[1] James Walters,[1] Simon Murphy,[3] Graham Moore,[3] Frances Rice,[1] Stephan Collishaw[1]

For numbered affiliations see end of article.

**Correspondence to**
Dr Stephan Collishaw;
collishaws@cardiff.ac.uk

## ABSTRACT

**Objectives** To coproduce a school-based protocol and examine acceptability and feasibility of collecting saliva samples for genetic studies from secondary/high school students for the purpose of mental health research.

**Design** Protocol coproduction and mixed-methods feasibility pilot.

**Setting** Secondary schools in Wales, UK.

**Participants** Students aged 11–13 years.

**Primary and secondary outcome measures** Coproduced research protocol including an interactive science workshop delivered in schools; school, parental and student recruitment rates; adherence to protocol and adverse events; ability to extract and genotype saliva samples; student enjoyment of the science workshop and qualitative analysis of teacher focus groups on acceptability and feasibility.

**Results** Five secondary schools participated in the coproduction phase, and three of these took part in the research study (eligible sample n=868 students). Four further schools were subsequently approached, but none participated. Parental opt-in consent was received from 98 parents (11.3% eligible sample), three parents (0.3%) actively refused and responses were not received for 767 (88.4%) parents. We obtained saliva samples plus consent for data linkage for 79 students. Only one sample was of insufficient quality to be genotyped. The science workshop received positive feedback from students. Feedback from teachers showed that undertaking research like this in schools is viewed as acceptable in principle, potentially feasible, but that there are important procedural barriers to be overcome. Key recommendations include establishing close working relationships between the research team and school classroom staff, together with improved methods for communicating with and engaging parents.

**Conclusions** There are major challenges to undertaking large-scale genetic mental health research in secondary schools. Such research may be acceptable in principle, and in practice DNA collected from saliva in classrooms is of sufficient quality. However, key challenges that must be overcome include ensuring representative recruitment of

### Strengths and limitations of this study

► This is the first study to test the feasibility and acceptability of collecting genetic samples in secondary schools and obtaining consent for linkage to questionnaire and record-based mental health data.
► A key strength is coproduction of the research protocol with stakeholders (young people, parents/guardians, schools).
► We used a mixed-methods approach to assess the feasibility and acceptability of carrying out genetic research studies of mental health in schools.
► This pilot study was conducted in three mainstream secondary schools in Wales, UK so it is unclear whether findings are transferrable to a wider section of schools in Wales and other countries, education systems and age groups.
► It was not possible to collect data on the reasons for return or non-return of parental consent.

schools and sufficient parental engagement where opt-in parental consent is required.

## INTRODUCTION

In the UK, approximately one in eight (12.8%) young people aged 5–19 years old have a diagnosable mental health disorder with rates increasing in recent years.[1 2] The causes of youth mental health difficulties involve genetic and environmental risk factors acting together in complex ways. The majority of adult mental health conditions originate before the age of 24,[3 4] and early identification and prevention are important priorities. However, only a minority of young people with mental health problems seek or receive help from healthcare professionals.[1 5] To better understand risk and protective factors

for psychiatric conditions, data from population-based samples of young people, including relevant genetic, biological, psychological and social factors are important. Established UK birth cohorts are a valuable resource for studying the development of mental ill health, including the interplay of genetic factors and family environment. However, the costs involved in setting up and maintaining such cohorts are considerable, and information about other social contexts such as schools is often limited.[6]

An alternative approach involves collecting data on mental health and associated risk and protective factors from young people within the school setting, offering the opportunity to study the roles of classroom, peer group and school-level effects. In addition, school-based designs offer the potential to recruit and obtain data from larger population-based samples than is possible using traditional birth cohort designs. Typically, student participation rates are high when health questionnaire data are collected during the school day.[7 8] What is unclear is whether it would be acceptable to schools, students and their parents to collect saliva samples for the purpose of genetic studies involving mental health, and what the main barriers are that need to be overcome to make this feasible in practice. Challenges include ensuring schools, parents/guardians and young people themselves will be accepting of research on genetics and mental health; providing information to young people, their parents and teachers; collecting appropriate informed consent; integrating research into the every-day life of schools in a way that fits with the needs of schools and learners and implementing robust and ethical protocols for the collection of saliva samples in a classroom setting.

Previous studies have had some success with collecting salivary cortisol samples in school settings (for reviews, see[9 10]). In contrast, little is known about the acceptability and feasibility of classroom-based collection of saliva samples for genetic research. Despite increasing understanding and acceptance of genetic research, public concerns remain—particularly in relation to children,[11–15] and mental health is often stigmatised,[16 17] so it is unclear whether this type of research would be acceptable to young people, parents/guardians and school staff. Similarly, the concept of data linkage (eg, to mental health questionnaires or health records) might elicit concerns about privacy.[18] Parent/guardian recruitment and consent is typically challenging in school-based research,[19–22] particularly in secondary school settings. Having a research study and protocol that is acceptable to key stakeholders is critical to a research study's success.[8 23 24] It will both help with recruitment, and will also help develop a process that key stakeholders engage with or 'buy into', and that fits with the context and daily life of students, teachers and parents. Indeed, coproduction of research with stakeholders is critical to support the development of school-based research.[8]

To the best of our knowledge, no other study has examined the acceptability and feasibility of collecting saliva samples from young people in schools for the purpose of genetic research on mental health.

## The current study
The Mental Wellbeing in Adolescence: Genes and Environment Study (MAGES) aimed to assess the acceptability and feasibility of collecting DNA saliva samples from young people in schools with consent for linkage to other routinely collected mental health questionnaire and record-based data. The over-arching aims were to work with stakeholders (school staff, parents and young people) to coproduce an acceptable research protocol, and then test this protocol in order to inform future studies on the best ways to carry out this kind of research.

The study was conducted in Wales which provides a globally unique research infrastructure, with student health, mental health and wellbeing data collected every 2 years in all mainstream secondary schools via SHRN (School Health Research Network, http://www.shrn.org.uk/) and potential linkage to routine health, education and social care data via SAIL (Secure Anonymised Information Linkage) databank (www.saildatabank.com). The SHRN 2017 health and wellbeing survey was completed by all state-funded schools in Wales, UK (n=193) and had 97% of students take part (n=1 12 045).[7 8]

In the development phase, we worked with stakeholders (young people, parents/guardians, schools) to develop a study protocol that had the greatest chance of being both acceptable and feasible in practice. To evaluate the MAGES protocol, we used a mixed-method design with quantitative and qualitative data. Specifically, we examined school, parent/guardian and student consent/participation rates, considered adherence to the study protocol and the occurrence of any adverse events (eg, complaints), and the ability to genotype samples. We collected feedback from young people and undertook focus groups with teachers to gain further insights on the feasibility and acceptability of the study, and how the protocol might be adapted in future.

## METHODS
### Study design
The study was conducted in three stages: first a development phase, followed by implementation of the protocol and then an evaluation phase (figure 1). The development phase included coproduction of the study protocol with key stakeholders. The MAGES protocol included recruitment of schools, obtaining consent from parents/guardians and students, and collection of saliva samples for genetic analysis. Saliva collection occurred during specially developed MAGES science workshops that took the place of a normal science lesson (see below). No phenotypic information was collected on participants. Quantitative evaluation included numbers and percentages for each stage of recruitment, per cent of usable genotyped samples, and student feedback scores on the science workshop aspect of the protocol. Qualitative

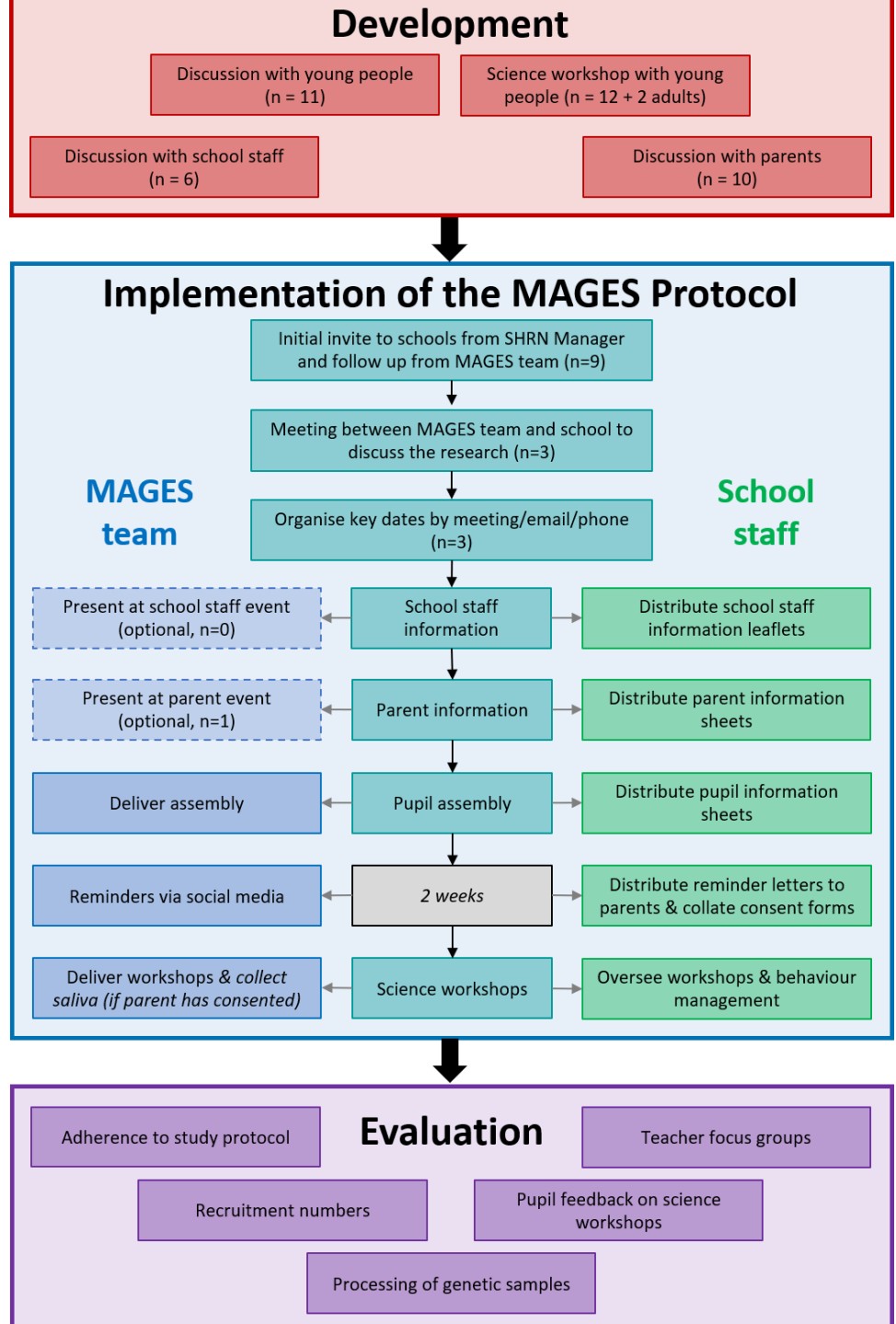

**Figure 1** MAGES recruitment and procedure. MAGES, Mental Wellbeing in Adolescence: Genes and Environment Study; SHRN, School Health Research Network.

evaluation included teacher focus group discussion of MAGES protocol following completion of classroom data collection.

### Development phase

Stakeholders involved during development of the research protocol included young people, school staff and parents/guardians.

Researchers discussed the study protocol and the practicalities of using saliva collection kits in a classroom setting with a group of young people aged 14–17 years old (n=11, 5 males, 6 females). Young people were part of the public patient involvement group ALPHA (https://decipher.uk.net/public-health-improvement-research-networks-phirns/public-involvement-alpha/). Based on feedback from this session, we made changes to the study

protocol (including a school assembly presentation) to simplify the content and to explain technical terms (eg, data linkage) more fully.

School staff shared their perspectives on the acceptability of taking saliva samples from students in schools and provided advice on practical issues. Teachers from nine schools that were engaged in SHRN research were invited to take part. A total of five teachers (three females, two males) from five schools and one Healthy Schools Practitioner (female) participated. Particular consideration was given to how research participation would impact teacher workload, how researchers could give back to schools and potential practical challenges. School staff highlighted that getting the parent/guardian consent required for participants aged under 16 years old (as is required in Wales, UK) was likely to be the most challenging aspect of the project. As a result of this session, we adapted our protocol to target younger year groups (years 7 and 8, age 11–13 years) as it was thought that parents/guardians would be more engaged and older cohorts could not afford to take time out of core lessons. Suggested ways to engage parents/guardians were to meet in person via events at each school, and by presenting MAGES information in different formats. We therefore included a parent/guardian event in our protocol and also created a website with videos explaining why the research is important and what taking part involves (www.cardiff.ac.uk/MAGES). Giving back to schools was also highlighted as important and providing a science workshop to students was considered a good way to do this.

Mothers (n=10) recruited from a local parent research network took part in a discussion on the proposed research and provided feedback on the clarity and content of parent/guardian information sheets. Data linkage emerged as a key concern and we adapted information sheets to provide more information on this.

Finally, to ensure that the science workshop content was suitable and enjoyable for the proposed age range, we trialled the science workshop (see below) with a local scout group of 12 boys aged 10–13 years old and two adult scout leaders (one male and one female).

### Implementation phase
#### Evaluation sample
Participants were students in years 7 and 8 (aged 11–13 years) at mainstream secondary schools in South Wales, UK that were part of the SHRN.[7 8]

#### Recruitment and protocol
Figure 1 depicts the recruitment and protocol used. First, schools that were consulted in the development of the protocol (n=5) were invited to take part in MAGES via direct correspondence from the SHRN Manager to each of the SHRN school contacts. This was followed up by MAGES staff. A further four local SHRN schools were invited to participate at a second recruitment wave. Participating schools were offered £500 (£250 per year group) as a thank you for facilitating the research and to cover costs in staff time resulting from participation.

*School staff meetings:* Following initial contact, MAGES researchers met with members of each school's senior leadership teams. All schools were given the option of holding events for parents/guardians and teachers where MAGES staff would introduce the project and answer questions.

*Information packs:* Schools were asked to disseminate parent/guardian information packs (using typical communication methods). These included an overview of the study, frequently asked questions, and a link to the study webpage (www.cardiff.ac.uk/MAGES). Parents/guardians were also given email and phone contact details for the MAGES team if they had queries or concerns.

At a later date, MAGES researchers delivered 15–20 min assemblies to students to explain the project, following which, the schools were asked to distribute information packs to students.

*Workshops:* Feedback from stakeholders during the development phase indicated the value of science workshops on the theme of genetics for engaging schools and learners. Student science workshops were scheduled to start 2 weeks from the student assembly. During this period, schools distributed reminder letters to parents/guardians and collated consent forms. MAGES staff also provided reminders via social media (twitter).

MAGES researchers delivered the workshop to all classes in each participating year group in the place of a normal science lesson (lasting 50–60 min). Science workshops began with an introduction to MAGES and the team followed by an interactive lesson (see figure 2) consisting of (1) a presentation teaching the basics of DNA, (2) a practical experiment extracting DNA from bananas, (3) an additional presentation on DNA structure, heredity, traits influenced by genes and impact of environment/experience, as well as an interactive discussion on non-visible traits that might be related to variation in base pair sequences (this was directed by student responses but often covered traits such as mental health, IQ, talents in sports and music and personality) and (4) an activity creating origami DNA models. During the origami activity, those students who had completed parent/guardian consent forms were invited to take part in the DNA collection. Students were given their own assent form to sign and then provided a saliva sample. This was conducted in a screened off area of the classroom or in a side room to provide privacy. At the end of the science workshop, all students were asked to provide feedback about whether they had enjoyed the science session on a sticker chart (online supplemental figure 1). Science workshops and data collection occurred between April and July 2019.

### Evaluation phase
#### Feasibility
*Recruitment and participants*
The numbers of schools recruited, parent/guardian consent forms returned, student participation and consent for data linkage were recorded and percentages of the eligible sample were calculated. Where possible, reasons for not taking part were recorded. To assess school-level response bias, participating and non-participating schools

## 1. Interactive presentation

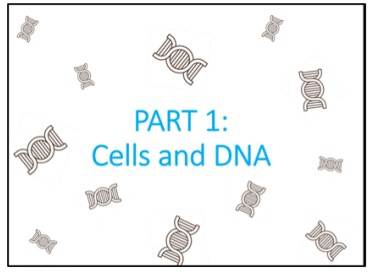

- All living things are made up of cells
- Cells are the building blocks of life
- Cell structure
- DNA is in the nucleus
- DNA is like a recipe book for your cells

## 2. Practical experiment extracting DNA from bananas

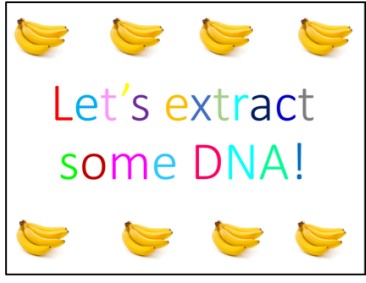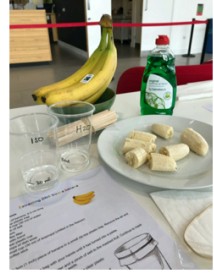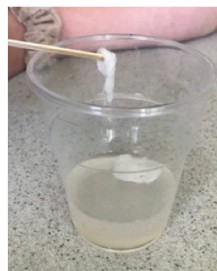

## 3. Interactive presentation

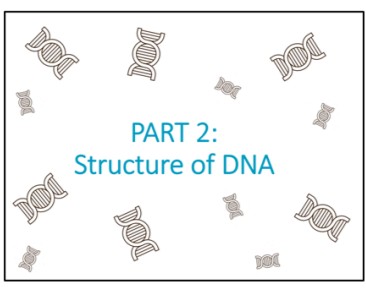

- Discovery of DNA structure
- DNA backbone and base pairs
- Genes and heredity
- External and internal traits influenced by genes
- Impact of environment and experience

## 4. Origami DNA model making

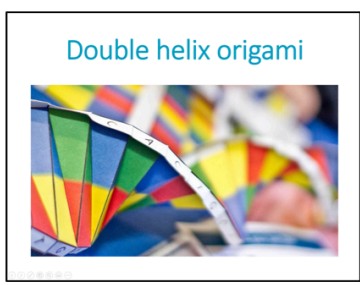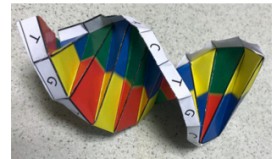

www.yourgenome.org/activities/origami-dna

**Figure 2** Science workshop structure and activities.

were compared on a number of routinely assessed school-level characteristics (https://mylocalschool.gov.wales/), including Free School Meals (FSM) entitlement (%), minority ethnic pupils (%), student attendance (%) and academic achievement (% achieving 5 General Certificate of Secondary Education at A*-C grades).

### Saliva samples

Participants provided saliva samples (approximately 5 mL) using Genotek Oragene saliva kits under the supervision of MAGES researchers (full instructions in online supplemental figure 2). Participants were asked if they had eaten or drunk anything in the last 30 min and if not, were instructed to fill the saliva collection tube to the fill line. If participants had eaten or drunk in the last 30 min, they were asked to wait 30 min before providing a sample. Sample collection took around 5–10 min per participant, and multiple students provided samples at the same time under researcher supervision. The collection tubes were labelled using barcodes and a unique participant study number.

The samples were taken to the research laboratory—MRC Centre for Neuropsychiatric Genetics and Genomics at Cardiff University. All samples were processed in accordance with the standard operating procedures for sample management, storage and tracking of biological materials. DNA was extracted from the saliva samples in-house, following standard Genotek Oragene DNA Prep-IT protocols. DNA sample quantification was determined using Quant-iT PicoGreen dsDNA assay kits, and samples were genotyped using Illumina Infinium Global Screening arrays. Data were recorded on the number and percentage of successfully extracted and genotyped samples.

### Adherence to study protocol and adverse events

The research team undertook a review of the protocol following completion of the study within each school and recorded data on adherence to protocol. This included instances where the protocol (figure 1) was changed and any adverse events (eg, complaints).

### Acceptability
### Science workshop

Student feedback was collected at the end of each workshop to assess the value of including the science workshops in the protocol. Students rated their enjoyment of the workshop using a sticker chart (online supplemental figure 1) with a scale of: (1) 'Yes—I had great fun'; (2) 'Most of it was quite good'; (3) 'Some of the time it was ok'; or (4) 'No—I didn't like it'.

### Teacher focus groups post-MAGES

Three focus groups were held with teachers in participating schools to get feedback on MAGES. Teachers were recruited to the focus groups by each school's key contact teacher. A £20 voucher was offered as remuneration for each teacher's time, and schools were given £125 for holding the focus group (to cover replacement teaching time). Five teachers participated in each focus group (school 1: three females, two males; school 2: five females; school 3: two females, three males). This sample included science teachers, members of the senior leadership team and form tutors responsible for pastoral care. Data were collected at participants schools at a time and date convenient to them.

Focus groups lasted approximately 45 min and were conducted by two female MAGES researchers (SR, Med, 3 years' experience of conducting and analysing focus groups; and NW, PhD, with training in semi-structured clinical interviews). Researchers were responsible for workshop delivery and saliva collection, and therefore had working relationships with the teachers prior to the focus group. Teachers were asked about their views on mental health research in young people, how MAGES was conducted in their school, how they and others (parents/guardians, students) found the MAGES process and what improvements they would make to the study (see online supplemental table 1 for focus group schedule). Digital audio recordings of the three focus groups were transcribed verbatim by a professional transcription company and supplementary handwritten notes were made. The transcribed interviews were then exported to NVivo V.12, a qualitative data analysis computer software package.

Two researchers (SR and NW) conducted an inductive thematic analysis of the data following Braun and Clarke's framework.[25] The steps in this process included: (1) data familiarisation, (2) initial code generation, (3) theme identification and framework development, (4) theme review and (5) final theme definition. A wide range of views were collected and researchers were confident that there was no further information that could have been gained from recruiting more participants/holding more focus groups.

Both researchers coded all the transcripts independently and then met to jointly develop a coding framework. This framework was derived inductively from the focus group data but was also influenced deductively by the research questions. They subsequently recoded the transcripts using the agreed framework using NVivo 12.

### Patient and public involvement

The design of the protocol was informed by extensive PPI work with key stakeholders—young people, parents/guardians and school staff (see 'Development phase' above). Schools participating in the development phase of the project were offered the opportunity to take part in the main study and help recruit parents and students at their school. Results were disseminated to participating schools through electronic and paper feedback reports.

## RESULTS
### Feasibility
### Recruitment and participants

Three of five schools involved in the advisory stage agreed to take part in MAGES, with the two non-participating schools stating they were too busy. No schools (0/4) in the second recruitment wave agreed to meet to discuss taking part in MAGES. Researchers were unable to reach the SHRN contact prior to the end of the study in two schools, and one schools declined taking part due to being too busy, and another due to concerns over taking DNA from children and being perceived as 'having young people with mental health problems'. The total school participation rate was 33.3% (three out of nine invited schools). On average, the three participating schools had lower FSM entitlement (14.0% vs 23.1%), lower proportion of minority ethnic students (15.4% vs 21.9%) than the six non-participating schools, and similar student attendance (94.3% vs 93.7%) and academic achievement (58.4% vs 60.2% students 5 GCSE A*-C grades). In comparison to the national average, participating schools had lower free school meal (FSM) entitlement (Wales average 17.5%), higher proportion of minority ethnic students (Wales average 9.8%), higher student

**Table 1** Participation and consent rates

| School (eligible sample) | Parent/guardian consent forms returned, n (%) | Student saliva samples, n (%) | Routine data sets linkage, N (%) | School questionnaire data linkage, N (%) | Routine data sets and school questionnaire linkage, N (%) |
|---|---|---|---|---|---|
| School 1 (n=246) | 34 (13.82) | 31 (12.60) | 29 (11.79) | 30 (12.20) | 29 (11.79) |
| School 2 (n=157) | 27 (17.20) | 24 (15.29) | 24 (15.29) | 21 (13.38) | 21 (13.38) |
| School 3 (n=465) | 37 (7.96) | 35 (7.53) | 35 (7.53) | 29 (6.24) | 29 (6.24) |
| Total (n=868) | 98 (11.29) | 90 (10.37) | 88 (10.14) | 80 (9.22) | 79 (9.10) |

Note: % of eligible sample.

attendance (Wales average 93.9%) and higher academic achievement (Wales average 55.1%).

Table 1 details the number of parent/guardian consent forms received, saliva samples collected and consent for data linkage for each participating school. Three parents from the eligible sample of 868 (0.3%) refused permission for their child to participate, either via email (n=1) or on the consent forms (n=2). Ninety-eight parents (11.3%) provided signed consent for students to participate in the study. No responses were received from the remaining eligible sample (88.4%).

Of the 98 students with parent/guardian consent, saliva samples were obtained from 90 students (89.6%; 31 males, 59 females). Five students decided they did not want to take part, two were absent on days of saliva collection and there was not enough time to collect a sample from one student. Consent for complete data linkage was obtained for 79 (80.6%) students.

Sample collection rates varied by school, ranging from 7.5% to 15.3% of eligible students. This primarily reflected variation in parent/guardian consent (8.0%–17.2%). There was also considerable within-school variation in sample collection between different classes (school 1: 0%–38.2%; school 2: 11.1%–28.0%; school 3: 0%–21.4%).

### DNA extraction and genotyping
We were able to extract DNA and genotype 89 of the 90 (98.9%) samples collected. One sample was not genotyped due to insufficient concentration of DNA.

### Adherence to study protocol
The study protocol was followed for school 1. However, the time-limited nature of science workshops during a normal lesson restricted the number of saliva samples that could be collected. In subsequent schools, we adjusted the protocol so that the saliva collection occurred approximately 1 week following the science workshop to allow adequate time.

After school 1 layout and formatting changes were made to the parent/guardian consent forms to increase clarity.

Only one school (school 3) opted to provide an event to explain MAGES to parents and guardians. This event was organised specifically to discuss MAGES (Thursday evening, 5.30 pm start) but was poorly attended (n=5, 1.1% of school eligible sample). This session was primarily comprised of parents and guardians with concerns and queries about the research. No school chose to have the additional event for MAGES researchers to explain the project to teachers.

We did not receive any complaints about the research from students, parents/guardians or school staff.

### Acceptability
#### Science workshop
Of the students who gave feedback on the science workshops, the majority (88.4%) said 'Yes—I had great fun' or 'Most of it was quite good' (table 2).

#### Teacher focus groups post-MAGES
A number of themes were identified from thematic analysis of qualitative interviews with teachers (table 3). Here we focus on key themes that informed our understanding of acceptability and feasibility.

#### Acceptability
Teachers were asked about their views on the appropriateness of conducting a study like MAGES in a school environment. Generally, teachers who were interviewed were in favour of such research.

> You asked whether or not it's a good idea to use the schools. I think we're in an ideal position. A captive audience, if you want. It's the easiest way of getting hold of those pupils and that information and of youngsters so I don't necessarily have a problem with schools being involved.

The consensus was largely that the MAGES protocol was acceptable, however there was some concern that this view may not be shared by other people within the community. Some teachers suggested that people outside of the school may feel it was inappropriate for teachers to facilitate this kind of research.

> I wonder how that might be seen by different people as in, why are they taking DNA? What are they going to do with it? Why should teachers allow them to come in and do that?

**Table 2** Student feedback on science workshops

| | Have you enjoyed the MAGES science workshop? | | | |
|---|---|---|---|---|
| | Yes—I had great fun, n (%) | Most of it was quite good, n (%) | Some of the time it was ok, n (%) | No—I did not like it, n (%) |
| School 1 (n=191) | 148 (77.49) | 27 (14.14) | 12 (6.28) | 4 (2.09) |
| School 2 (n=119) | 79 (66.39) | 30 (25.21) | 5 (4.20) | 5 (4.20) |
| School 3 (n=343) | 185 (53.94) | 108 (31.49) | 34 (9.91) | 16 (4.66) |
| Total (n=653) | 412 (63.09) | 165 (25.27) | 51 (7.81) | 25 (3.83) |

n reflects the number of students present in class who chose to give feedback on the MAGES science workshop.
MAGES, Mental Wellbeing in Adolescence: Genes and Environment Study.

Teachers discussed the acceptability of MAGES from the point of view of parents/guardians and students. Although some participants suggested that they expected parents/guardians to react negatively to MAGES, all participants agreed that no parent or student approached them with any complaints or concerns.

When I first got sent the email about the project, as a scientist I thought some parents are not going to like that…but we took the risk and, in fact, we got more people coming back than I thought we would.

### Benefits

The benefits of taking part in MAGES were widely discussed, with members of all groups indicating that they would be willing to participate in MAGES again in the future. The potential contribution to mental health research was noted in all three focus groups as a major benefit of being involved with MAGES.

"I think there's a lot of mental wellbeing issues in amongst children now. If we've got research and there's data on it, if that data can be used in a positive way, then it's a good thing but it's just the feasibility of collecting that large amount of data for it to be viable."

Teachers also said they would have agreed to take part without the incentive of the science workshop, however there was a preference for the workshop to remain as part of MAGES.

"I would have still agreed to do it, absolutely, but I wonder if the kids could actually link to what's going on. I think that's where the disconnect would be. We still would've signed up to it absolutely because we recognise we've got mental health issues in the school and the importance of these types of research studies."

Similarly, teachers acknowledged the value of linking genetic information with data on mental health.

"I would've thought, to make your research valuable, you've got to do it otherwise all you've got it is a DNA sample."

The possibility that genetic research may become more acceptable to people in the future was brought up by multiple teachers.

"I think that attitude will change in the future. This is quite early on. Everybody was initially technology—the beast. Now, everybody's embracing it. I think exactly the same thing will happen with DNA and testing. I think it will probably become quite routine."

The prestige of working in partnership and forming a relationship with Cardiff University was also seen as a benefit.

"Our incentive has been the formation of this partnership and feeling like we're helping you with your samples and we've had something for our students back."

### Science workshop

The biggest benefit identified was the science workshop that was delivered to all year 7 and 8 students. Teachers frequently commented on the value of having external visitors who could be viewed as role models. Science workshops were seen as helpful to clarify how students' saliva samples would be used if they chose to take part. Teachers noted how much students enjoyed the session and suggested it gave them an opportunity to practice real, advanced science relevant to the teaching curriculum.

"It creates a buzz that you've got the profile of Cardiff University coming in, it gets the children excited about it. When we have outside speakers, they love that and that's why I think it's needed because otherwise, it's just spit into this thing and signing a form whereas if they did the banana DNA, they went home and talked about it, they were talking about it in their next lessons."

"I think it's really important for students to see researchers because students have this idea that scientists are lab coats and you don't all look like Albert Einstein. For you guys to come in, you're normal people and to say 'we are scientists, we are doing this' and for them to think, 'you're ordinary people, we could do that.'

**Table 3** Themes identified from qualitative analysis of teacher focus groups

| Main theme | Subtheme | Second subtheme |
|---|---|---|
| **Acceptability** | Value of data-linkage | |
| | More acceptability in the future | |
| | No expressed concerns | |
| | Perception of parent/guardian and child acceptability | |
| | School would take part again | *Would take part without workshop* |
| | Using schools for genetic research | |
| **Benefits** | Partnership with CU | |
| | Mental health research | |
| | Science workshops | *Advanced science* *Benefits research* *Student enjoyment* *External visitors* *Real science* *Role models* *Useful for teaching* |
| **Concerns** | Children do not understand | |
| | Future use and impact | |
| | Linkage—data privacy and access | |
| | Parent/guardian concern of genetics | |
| | Perceptions of mental health testing | |
| | Potential harm to participants | *Determinism* *Finding out* |
| **Challenges** | Communication with parents/guardians | *School contact with parents/guardians* *Lack of parent/ guardian understanding* |
| | Communication with students | |
| | Communication with teachers | |
| | Recruitment | |
| **Suggestions for the future** | Engaging parents/ guardians | *More information* *Parent event* |
| | Engaging students | *Assembly* *Enthusiasm* *Science workshops* |

Continued

**Table 3** Continued

| Main theme | Subtheme | Second subtheme |
|---|---|---|
| | Working with teachers | *Engagement* *Involving form tutors* *Involving all staff* *Science department* |
| **Large scale MAGES** | Logistics | |
| | School variability | |
| | Views on expanding study | |
| | Workshop going forward | |
| **Mental health** | Awareness | |
| | In schools | |
| **Practicalities** | Disseminating MAGES information | |
| | Teacher workload | |
| | Timing and organisation | |

CU, Cardiff University; MAGES, Mental Wellbeing in Adolescence: Genes and Environment Study.

## CONCERNS AND POTENTIAL CHALLENGES

Although focus group participants agreed on the whole that it was acceptable to conduct MAGES within a school environment, they did acknowledge some concerns. These focused on the potential negative impact of genetic research on participants as well as privacy issues surrounding the process of data linkage.

All potential MAGES participants were told that they would not receive any results from their saliva sample during the initial assembly, in the science workshop and in all MAGES information packs. However, not all teachers were present in the assembly and many had not read the information leaflets. This led to some unaddressed concern among teachers about the potential harm that could be caused to students if they were to be informed that they had an increased genetic risk for particular mental health conditions. Focus group participants felt strongly that students should not receive feedback regarding the results of their DNA sample as feedback might lead students to believe they were predisposed to mental illness.

"If we've got a young person who has mental health issues, they get a DNA test, they find they've got that gene, I fear they'd think there's nothing they could do. They'd say, 'I've got the gene, I'm genetically going to have mental health issues, there's no point having therapy, there's no point talking about it because that's just who I am.' "

Concerns about how participants' data could be used in the future and the potential negative impact this might have were discussed.

"If you discover a DNA precursor to mental health, what if an insurance company in the future said to you is this person likely to get mental health illnesses? Or a mortgage company?"

Teachers acknowledged the value of data linkage and were aware of the measures in place to protect participant's privacy, however some did still express concern.

"I know its numbers and barcodes but at some stage in the process, somebody will have access to the names and be able to link it.

Participant understanding of MAGES and communication were noted as challenges. Some teachers felt that the information given to students by researchers about genetic research and data linkage was too complex for young people to properly understand.

"It's the lost in translation thing—they didn't quite understand … and when the kids are very, very weak [academically], it was more lost in translation.

The initial MAGES assembly was felt to be too complex and that this had led to misunderstanding the purpose of the study by some students. Teachers said that some students came away from the assembly believing that the purpose of MAGES was to screen them for mental health conditions.

I had one student …who thought you were going to test her for mental health problems and was concerned that you were going to tell her there was something wrong with her."

Similarly, teachers suggested that parents/guardians may have found the information sheets to be too complex which may have impacted their decision about allowing their child to participate.

"It's education for the parents as well—they need to fully and truly understand what it's for, what's happening to their child's DNA what are they going to do with it, what's going to happen in the end? Obviously, we do have a lot of parents who … don't truly understand what it means to take DNA and they just understand DNA from the television… If they don't truly understand why you're taking it then no, it's too scary…"

Communication, in particular, was seen as a challenging element of MAGES, and that this required teachers to provide additional information and answer follow-up questions from students. Teachers reported that some staff members were approached by students with questions about MAGES following the initial assembly and the distribution of the student information sheets, suggesting that the information provided by researchers was inadequate on its own.

"When I gave out the packs, I asked if there were any questions and I spent 10–15 minutes with people asking if it will tell them if they've got this disease and will they have this on file forever."

Teachers felt that not enough school staff were given information about MAGES and that this limited the school's ability to facilitate the recruitment of potential participants.

"They [students] would come and ask me, some of them who are in my class, but I think because the other science teachers weren't massively, well they didn't really know what this was about and what was going on, perhaps they weren't as enthusiastic as I was."

The majority of teachers felt that the school's contact with parents/guardians regarding MAGES was ineffective which may have had negative implications for recruitment.

Parents—we didn't get them in … the only way we managed to get it out was on our 'Schoop Line' and via the letters. So, it was woeful in that respect in terms of engaging the parents.

## Recommendations for the future
### Working with school staff

Focus group members suggested various ways for researchers to more effectively engage staff in the participating schools. This included involving more staff throughout the school including science teachers, school nurses and teachers responsible for pastoral care. Participants suggested that the most effective way to engage with school staff would be for MAGES researchers to organise a face-to-face meeting to present information verbally.

"I wonder as well… if because I spoke to Year 7 and 8 tutors only. I've mentioned it to other staff but in passing. I wonder if every member of staff in the school community could be aware of what is going on."

### Engaging students and parents/guardians

There was significant discussion of the importance of MAGES researchers engaging with students and suggestions of several ways in which this could be improved. Proposed improvements included simplifying the initial student assembly, making MAGES more exciting and appealing to students and alternative DNA-related activities that may be more relevant to the research.

"I feel that maybe it could have been sold as a bit more fun and special as in you're helping people out, you're doing this, not everyone's getting to do it. Because you had to say all the important bits and everything ethically, that then it didn't seem as fun for them…You've got to give the information but I'm wondering if it could be sparkled up."

As parental consent was a necessary prerequisite for student participation in MAGES, this was discussed extensively in focus groups as a key area in which to boost recruitment. A parent event in which MAGES researchers meet face-to-face with parents/guardians to answer

questions and provide detailed information was considered to be the most effective way to achieve this.

> "I think that if we were to do this again, then we would look to hold an evening for parents, as everybody has said, to get the elephant out of the room and have those discussions."

## DISCUSSION

This study aimed to develop and test a protocol to obtain genetic samples in schools for mental health research. While genetic and mental health research was viewed as important and acceptable by stakeholders in the development phase, and the protocol itself proved largely acceptable, we also found that the protocol was not feasible in its current form due to a number of challenges, notably non-response from parents and securing school participation. This protocol was highly resource-intensive, and further consideration of resources is required to make the protocol more effective if data collection is to be scaled up. The quality of saliva samples was good with only one sample unable to be genotyped, which suggests researcher-supervised saliva collection using spit kits is a viable method of collecting genetic data from young people in schools. We received no complaints from students, parents or school staff concerning the study and only three active refusals from parents at the consent stage. The MAGES science workshops were viewed as an important (but perhaps not essential) component by teachers, and received positive feedback from the majority of students. Teachers saw mental health as important, and were, in principle, accepting of collecting genetic data for the purpose of mental health research in schools; however, this information is limited to teachers from schools that took part, therefore were already interested and invested in such research. Teachers also highlighted concerns and challenges, such as improving communication and engagement, that should be addressed going forward.

A major strength of this study is the inclusion of stakeholders throughout the research process—from development through to evaluation. This allowed us to coproduce a study protocol with schools, young people and parents. Notably, the majority of schools who had participated in the coproduction phase participated in MAGES, compared with none of the schools contacted subsequently. We took a mixed-methods approach, giving more depth of information than just quantitative or qualitative research alone. We were also able to increase awareness of mental health and genetics among stakeholders especially young people which, although not our primary aim, has been a positive outcome of the study.

Nevertheless, this study has limitations. While parents and guardians were involved in the development phase, further information is needed to understand barriers to parent/guardian recruitment. At the individual level, the biggest driver of non-participation was parents not returning consent for their children's participation (rather than active refusal by parents or withdrawal by children). We were unable to contact parents directly so we were unable to collect information from parents regarding whether they had received information about the study and their reasons for not giving consent. We also did not collect phenotypic data on our participants so were unable to test predictors of non-participation directly. Engagement of parents/guardians can often be an important barrier to recruitment.[19–22] The current study required opt-in parental consent but this requirement varies across countries. In future, it will be important to develop research protocols that allow direct communication with parents. The study is also limited by the small number of schools that took part, and the limited uptake of teacher and parent MAGES meetings in these schools. This study took place in mainstream schools in Wales (UK), so results may not generalise to different education systems, countries and age groups.

Another limitation is the lack of diversity in our sample. As this was a small study assessing feasibility, we recruited an opportunity sample which may not have fully covered the diverse set of contexts needed to inform decisions going forward. This is important because there is much research, across multiple study designs, demonstrating that there are important differences between participants and non-participants in mental health research, with notable predictors of response that include affluence, family adversity, gender, educational attainment, behavioural problems, mental health and elevated genetic risk for mental health and neurodevelopmental problems.[26–28] In our study, at a school level, factors related to non-response included eligibility for FSMs which was higher in non-participating schools and the national average, indicating a wealthy volunteer bias. In addition, although our participating schools had a greater proportion of ethnic minority students than the national average, they had a lower proportion compared with schools whom we approached but which chose not to participate. In future, it will be important to understand more about recruitment and retention in ethnically diverse populations and develop research protocols that ensure that traditionally underrepresented groups are closely involved in the coproduction of the research. This is particularly important for health-related research to ensure that research findings are relevant to marginalised groups who often have a high burden of mental health difficulties. It is also important so that policy and practice recommendations that follow from research are developed appropriately and fairly. We would argue that a coproduction approach to genetic mental health research is essential, and that a priority is to find ways to develop new research of this kind that deals explicitly with potential barriers to participation with input from marginalised groups from the outset.

A further important point is that, as this study was focused on feasibility to inform future decisions, the overall sample size was small. While the findings provide

helpful insights on the acceptability and feasibility of the methods used, it is critical to remember that data from much large numbers of individuals are essential for standard genomic analyses. The current approach was both labour and cost-intensive and it may be that broader scale awareness raising and social media campaigns may be more effective, such as those currently used to recruit participants to genetic mental health studies in adults.[29]

While the current study focused on evaluating a protocol to engage children in genetic mental health research in schools, it is also important to consider the role of the broader social and cultural context with acceptability of different approaches to genetic research also dependent on building public understanding and trust at a societal level. There is some evidence for a decline in trust over time[30] (with survey response rates showing a general decline),[31–33] as well as variation between countries in levels of public trust in science.[34]

To the best of our knowledge, this is the first study that has assessed in detail the feasibility and acceptability of collecting saliva samples in schools for the purpose of genetic studies together with obtaining consent for data linkage. However, parent consent in the current study was lower than other school-based research in other contexts and countries. For instance, in a school-based study collecting smoking survey information and genetic samples in 14–15 year-olds, there was a parental consent rate of 54%,[19] and a school-based survey study trialling recruitment methods in 6–7 year-olds obtained 56% parental consent in cohort 1 and 71% in cohort 2.[20] These studies were able to undertake more intensive recruitment strategies (eg, multiple waves of letters sent directly to parents, follow-up phone calls, incentives) over a longer period of time. The added complexity of linking genetic data to health records in the current study may have also affected response rates given concerns of confidentiality rank highly in reasons for parent consent refusal,[19] and teachers in post-MAGES focus groups highlighted genetic privacy as a concern. Teacher concerns were similar to those identified in previous research such as concerns about general privacy and the negative impact of potential future data disclosure (eg, insurance and mortgage company discrimination).[11 12]

Our study suggests that it is very difficult to reach a full cross-section of parents or for such work to be undertaken at scale or to be representative of the whole population. Family-based study designs such as population-based birth cohorts, or clinic-based recruitment of children with mental health conditions and their families appear better placed for engaging parents directly with biological sample collection, including genetics. This is particularly the case in circumstances where an effective link between the research team and the family has helped establish trust and mutual understanding, for example, as part of ongoing longitudinal population, patient or high-risk cohorts.

Our research points to a number of recommendations for future school-based mental health genetic research based on feedback from teachers and our own experience. First, engaging all stakeholders through the entire research process, from development to evaluation, is crucial. This not only facilitates recruitment and improves research protocols, but helps promote understanding of genetics and mental health among stakeholders, and the needs and perspectives of stakeholders among researchers. Face-to-face meetings are potentially best and should be included in school-based research protocols where possible, but this does have implications for researcher time and costs. Second, clear communication is essential for getting key messages to all stakeholders at all stages of the research. Factors that can aid clear communication are: simplified and concise information letters, multiple formats of information (eg, video messages, paper letters, website, face-to-face meetings), direct channels of communication by the study team to all stakeholders (one limitation of our study was that it was not possible to contact parents directly), and working with stakeholders to develop information packs and to introduce the research in schools. Third, it is important to give back to schools to reflect the time and hard work required to effectively facilitate such research. The science workshops in particular were highlighted as a major benefit for students and teachers and we also provided schools remuneration for their time. Again, scaling up would have significant cost implications. Fourth, adequate time and resources need to be dedicated to the collection of saliva samples. For instance, we altered the MAGES protocol to provide additional time for this. Finally, clear strategies for parent recruitment are needed for each school based on consultation with school staff. This is likely to include multiple waves of information packs sent direct to parent addresses, telephone follow-ups and providing multiple ways to make it as easy as possible for parents to consent (eg, paper form, electronic form by email, online forms).

Future research would benefit from investigation of how to enhance parental recruitment rates. Parental consent is a challenge in school-based research,[19–22] and may be particularly challenging in secondary schools compared with primary schools where parental links to schools are not as strong. Parental consent may also be particularly challenging with research that covers mental health, genetics and data linkage. Typically, large scale DNA collection has worked when parents have been present, for instance Spit for Science (https://lab.research.sickkids.ca/schachar/spit-for-science/), but this is not always possible in school settings. We chose to recruit younger students from UK secondary schools (aged 11–13 years) as consultation with key stakeholders suggested parents of this age group would be more engaged; however, our low parent recruitment numbers suggest this may not be the case. Research focusing specifically on factors that affect parental rates of opt-in consent for school-based studies of this kind is needed. It would also be beneficial to assess whether parents would be easier to reach and be more engaged at other stages in their children's school careers, for instance, parents of primary school aged children

(aged 4–11 years), though this would raise new questions about children's understanding and stakeholder views on the acceptability of genetic mental health research in this age group. Alternatively, research could focus on older students (eg, in the UK aged 16+ years) where participants are able to provide their own active consent; however, in practice this would not obviate the need to keep all stakeholders in the school community, including parents, appropriately informed about the purposes and practicalities of the research. The current research took place in 2019. In view of the ongoing challenges faced by schools in returning to face-to-face learning, COVID-related risk management and the additional pressures on delivering the core curriculum, it is likely that researchers will face additional challenges with recruitment of schools and in the engagement of parents/guardians if research of this kind were to be conducted now.

## CONCLUSIONS

Our study suggests that it is challenging to collect genetic data for the purpose of mental health research in a school setting. Low participation rates among parents indicate that the scope and scale of such research would likely be restricted to sample designs where it is less important that samples are representative at a whole population level. Ultimately, large-scale representative samples covering a broad spectrum of genetic, biological, psychological and social factors are required for advancements of our understanding of mental health risk and resilience in young people. The current study highlights that there would be major challenges in scaling up school-based mental health genetics research. The most important barrier is the difficulty in obtaining parent/guardian opt-in consent for their child's participation.

**Author affiliations**
¹MRC Centre for Neuropsychiatric Genetics and Genomics, Cardiff University, Cardiff, UK
²Population Health Sciences, University of Bristol Medical School, Bristol, UK
³Centre for Development, Evaluation, Complexity and Implementation in Public Health Improvement, Cardiff University, Cardiff, UK
⁴School of Psychology, Cardiff University, Cardiff, UK

**Acknowledgements** We are extremely grateful to the school staff, parents and young people involved in the protocol development and the students who took part in the study. We would also like to thank Joan Roberts for her assistance liaising with schools, Pete Gee for help with liaising with PPI groups and the fieldworkers who assisted with science workshops in schools (Silvia Colonna, Catherine Ollerhead, Eleanor Horton, Bethany Butler, Victoria Powell, Henry Breary and Morgan Moriarty).

**Contributors** NW contributed to the design of the study, data collection, conducted the quantitative and qualitative analysis, drafted the manuscript and gave final approval of the version to be published. SR contributed to design of the study, data collection, conducted the qualitative analysis, critically revised the manuscript for important intellectual content and gave final approval of the version to be published. RBJ and RB contributed to design of the study, contributed to the qualitative analysis, critically revised the manuscript for important intellectual content and gave final approval of the version to be published. LB, LH-J and AE contributed to design of the study, managed the storage and analysis of genetic samples, critically revised the manuscript for important intellectual content and gave final approval

of the version to be published. JH, KL, AT and JW contributed to conception of the study, obtaining funding, critically revised the manuscript for important intellectual content and gave final approval of the version to be published. SM, GM, FR and SC contributed to conception and design of the study, obtaining funding, critically revised the manuscript for important intellectual content and gave final approval of the version to be published. All authors agreed to be accountable for all aspects of the work in ensuring that questions related to the accuracy or integrity of any part of the work are appropriately investigated and resolved. SC is the guarantor and accepts full responsibility for the work and the conduct of the study, had access to the data, and controlled the decision to publish.

**Funding** This work was supported by a UK Medical Research Council Mental Health Data Pathfinder grant (MC_PC_17212). The work was undertaken with the support of The Centre for the Development and Evaluation of Complex Interventions for Public Health Improvement (DECIPHer), a UKCRC Public Health Research Centre of Excellence. Joint funding (MR/KO232331/1) from the British Heart Foundation, Cancer Research UK, Economic and Social Research Council, Medical Research Council, the Welsh Government and the Wellcome Trust, under the auspices of the UK Clinical Research Collaboration, is gratefully acknowledged. The work was undertaken with support from the Public Health Improvement Research Network (PHIRN). PHIRN is part of the National Centre for Population Health and Wellbeing Research funded by Health and Care Research Wales, Welsh Government http://www.healthandcareresearch.gov.wales/

**Competing interests** None declared.

**Patient consent for publication** Not required.

**Ethics approval** The study was approved by Cardiff University School of Medicine Research Ethics Committee (ref 18/57). As students were under 16 years, participation in MAGES required informed parental opt-in consent and student assent. Both parents/guardians and students had the option to provide or not provide additional consent/assent for linking genetic information to other routinely collected data. Consent for routinely collected data was split into two broad categories: (i) health and educational records, and (ii) student-completed health and well-being questionnaires.

**Provenance and peer review** Not commissioned; externally peer reviewed.

**Data availability statement** Data are available upon reasonable request from the corresponding author.

**ORCID iD**
Naomi Warne http://orcid.org/0000-0001-6160-8982

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
