## [Reviewer comments · BMJ Open]

ARTICLE DETAILS

TITLE (PROVISIONAL)	Collecting genetic samples and linked mental health data from adolescents in schools: Protocol co-production and a mixed-methods pilot of feasibility and acceptability
AUTHORS	Warne, Naomi; Rook, Sarah; Bevan Jones, R; Brown, Rachel; Bates, Lesley; Hopkins-Jones, Lucinda; Evans, Alexandra; Hall, Jeremy; Langlely, Kate; Thapar, Anita; Walters, James; Murphy, Simon; Moore, Graham; Rice, F; Collishaw, Stephan

VERSION 1 – REVIEW

REVIEWER	FERNANDEZ, Arnaud Hôpitaux Pédiatriques de Nice CHU-LENVAL, Child and adolescent psychiatry
REVIEW RETURNED	19-Mar-2021

GENERAL COMMENTS	I had the great pleasure of reviewing this study which aimed to develop and test a protocol for obtaining genetic samples in schools for mental health research. The whole thing is clear, well written (my level of English does not allow me to judge this aspect any further). The article is innovative, addresses a major public health issue (adolescent mental health) and highlights the major challenges ahead in the field. This study highlights that it is still difficult to conduct research from researchers to the general population and that it will be necessary in the future to actively involve the general population in the design of research. In this sense, this article has an appreciable philosophical dimension and more work of this type should be carried out to allow for innovation in the links between the public and researchers. I have no questions, but I would like to know more about next steps of the MAGES project after this work, if possible.
--

REVIEWER	von Schantz, Malcolm University of Surrey, Faculty of Health and Medical Science
REVIEW RETURNED	18-May-2021

GENERAL COMMENTS	This is a generally well written presentation of the co-production process of a project together with participating Schools. I think that clarification on the following points would be helpful:
---

	1. The workshop clearly provided theoretical and practical insights into DNA. Did it also include any actual elements of how genetic variation relates to behaviour in general and mental health in particular? 2. Similarly, I am not clear over whether, in addition to the DNA samples, any phenotypic information was actually collected from the participants. 3. One envisions an elephant in this classroom that I do not see mentioned in the manuscript is whether the execution project was affected by the Covid-19 pandemic, and especially, if there are any reasons to assume that the disappointing response rate from parents/guardians could relate to this.
--	--

REVIEWER	Rayner, Christopher King's College London, Institute of Psychiatry, Psychology & Neuroscience
REVIEW RETURNED	05-Oct-2021

GENERAL COMMENTS	This is a very well written protocol and pilot study outlining the authors' approach towards collecting genetic and mental health data from adolescents in schools. I recognise the difficulties with recruiting participants into such studies, especially where mental health and genomics are the key focus. This paper is a great example of how different approaches can be applied to engage and recruit participants, as well as the current limitations and obstacles to achieving these objectives. Below, I have noted a few thoughts and questions that came to mind while reading the manuscript. I hope the authors will find them useful. I wondered if the paper might benefit from further discussion of the differences between participating and non-participating schools. For example, the authors found that there was lower FSM in participating schools when compared to non-participating schools and the national average. Healthy-wealthy volunteer bias is common in research studies and warrants discussion here. Additionally, I thought it might be worth clarifying where point estimates are statistically different between the comparison groups? The lower proportion of minority ethnic students in participating schools could reflect lower levels of trust in scientists and health professionals in individuals from these groups - for which there is plenty of literature - perhaps also worthy of discussion here. I assume that the differences in the distributions of student ethnicity between the participating schools and the national average are because participating schools were also more likely to be urban rather than rural schools, which is also unremarked upon. Was there any data on the ethnicity of the participating students for each school? Is it not also a limitation that ethnic minority students are grouped into a single category? Was there more detailed information on students' backgrounds? To gain the trust of underrepresented groups, should researchers make more effort to recognise specific racialised groups, as well as their needs and concerns? How does racism impact on willingness to participate in studies? How does racial abuse impact mental health? A stronger
---

	argument on the need for representative samples could have been achieved in the manuscript. Data from underrepresented groups are essential to ensure that healthcare interventions from such research benefit everyone equally. As such, further discussion is required on the need for representative samples, and the challenges that researchers face in recruiting families from ethnic minority backgrounds and those with less socioeconomically advantaged backgrounds. Given that this is a common limitation across several British cohort studies, I wondered whether these findings could be compared with previous findings. Much of the issue of participation also hinges on the public trust of scientists and governments, to handle their data responsibly and use it for the right purposes. How can researchers use education to increase levels of understanding and trust? Are there examples in other countries that we can learn from? Related to this, I wondered how the science workshop and various education materials were developed and designed? Were public engagement specialists consulted? Would this help? How could a larger audience be engaged with educational material? I wondered what could be learned from TV shows such as “Educating...”, “The secret life of 4 and 5-year-olds” and the works of science communicators like Dr Adam Rutherford and Dr Hannah Fry in communicating the themes highlighted in this manuscript to the public. Given the enormous costs of mental health on society due to its effect on educational, occupational and life outcomes, perhaps TV and radio media could be produced to communicate these issues to a wider audience. Previous studies like the GLAD study have employed social media campaigns and news interviews to increase awareness and recruit large numbers of individuals (>40,000), which are essential for genomic analyses. Perhaps a more thorough discussion of different approaches to overcome poor rates of participation would be beneficial for planning future initiatives. Sample sizes required for genomic analyses were not really discussed. What should be the ambition of such initiatives? How many participants are required for genomic analyses? Furthermore, if sufficient data were obtained, which analyses would be carried out? How could this data improve outcomes for school children? I wondered whether any analyses of the SHRN phenotypic data had been conducted? And what did it show? And subsequently, why is collecting genomic data the logical next step? Is it economically justifiable? Perhaps specific examples of the influence of genetics on mental health and educational outcomes in school-aged cohorts could be referenced in the introduction to justify this direction of research. I felt that some of the teachers' testimonials could have been paraphrased such that they are less critical, more sensitive, and less repetitive: e.g. “It’s the lost in translation thing – they didn’t quite understand what. And some of them are quite, we have got a very, very weak [academically] to begin with and when the kids are very, very weak [academically], it was more lost in translation.” could be: “It [gets] lost in translation, they didn’t quite understand
--	--

	what [the research involves]. Some of the students are ... weak[er] [academically] ... and [for them]... it was more lost in translation.” Also: “I had one student ... who thought you were going to test her for mental health problems and was concerned that you were going to tell her there was something wrong with her.” Typing error on Page 21, line 24: “Parental consent is a challenge in school-based research,(19–22) and may be particularly challenging in secondary schools compared to primary schools where parental links to schools are not as strong as well as with research that covers mental health, genetics and data linkage.”
--	--

VERSION 1 – AUTHOR RESPONSE

Reviewer 1 Comment 1: I had the great pleasure of reviewing this study which aimed to develop and test a protocol for obtaining genetic samples in schools for mental health research.

The whole thing is clear, well written (my level of English does not allow me to judge this aspect any further).

The article is innovative, addresses a major public health issue (adolescent mental health) and highlights the major challenges ahead in the field.

This study highlights that it is still difficult to conduct research from researchers to the general population and that it will be necessary in the future to actively involve the general population in the design of research. In this sense, this article has an appreciable philosophical dimension and more work of this type should be carried out to allow for innovation in the links between the public and researchers.

I have no questions, but I would like to know more about next steps of the MAGES project after this work, if possible.

Author response: Thank you for your comments. We are pleased that you enjoyed reading the manuscript.

The study was designed as pilot research to produce recommendations that might be useful for others developing similar research in the future. The MAGES project has now been completed, and the findings from this and other related projects are informing our ongoing research on youth mental health in schools.

Reviewer 2 comment 1: This is a generally well written presentation of the co-production process of a project together with participating Schools. I think that clarification on the following points would be helpful:

The workshop clearly provided theoretical and practical insights into DNA. Did it also include any actual elements of how genetic variation relates to behaviour in general and mental health in particular?

Author response: Thank you for your comments. The science workshop included a presentation on variation in base pair sequence and an interactive element (discussion) with students on non-visible traits that might be related to genetic variation. The discussion was directed by student responses but often covered traits such as mental health, IQ, talents in sports and music, and personality.

Reviewer 2 comment 2: Similarly, I am not clear over whether, in addition to the DNA samples, any phenotypic information was actually collected from the participants.

Author response: We did not collect any phenotypic information on participants. The purpose of the study was to test feasibility and acceptability of collecting genetic samples together with consent for linking to phenotypic data already available (health records and separately available health and wellbeing questionnaires). This has been clarified in the manuscript: "No phenotypic information was collected on participants." (page 6, lines 114-115)

Reviewer 2 comment 3: One envisions an elephant in this classroom that I do not see mentioned in the manuscript is whether the execution project was affected by the Covid-19 pandemic, and especially, if there are any reasons to assume that the disappointing response rate from parents/guardians could relate to this.

Author response: Data collection for this study occurred April 2019 to July 2019, before the COVID-19 pandemic. We have included this in the Methods section (page 8, lines 186-187).

Given that the pandemic would likely introduce more complexity to this kind of research we have considered this in the Discussion: "The current research took place in 2019. In view of the ongoing challenges faced by schools in returning to face-to-face learning, COVID-related risk management, and the additional pressures on delivering the core curriculum, it is likely that researchers will face additional challenges with recruitment of schools and in the engagement of parents/guardians if research of this kind were to be conducted now." (page 20, lines 602-606).

Reviewer 3 comment 1: This is a very well written protocol and pilot study outlining the authors' approach towards collecting genetic and mental health data from adolescents in schools. I recognise the difficulties with recruiting participants into such studies, especially where mental health and genomics are the key focus. This paper is a great example of how different approaches can be applied to engage and recruit participants, as well as the current limitations and obstacles to achieving these objectives. Below, I have noted a few thoughts and questions that came to mind while reading the manuscript. I hope the authors will find them useful.

I wondered if the paper might benefit from further discussion of the differences between participating and non-participating schools. For example, the authors found that there was lower FSM in participating schools when compared to non-participating schools and the national average. Healthy-wealthy volunteer bias is common in research studies and warrants discussion here. Additionally, I thought it might be worth clarifying where point estimates are statistically different between the comparison groups?

Author response: Thank you for your comments. We have added substantially to the discussion of participation bias, why this is an important issue, and what we might conclude from our study around the challenges in recruiting a representative sample of students using the kind of approach we piloted in our study (page 18, lines 509-528). Our study suggests that there may be important biases related both to predictors of school participation and achieving parental consent.

FSM entitlement was measured at the school level and we do not have access to individual level data. Consequently, we do not have adequate power to test differences statistically.

Reviewer 3 comment 2: The lower proportion of minority ethnic students in participating schools could reflect lower levels of trust in scientists and health professionals in individuals from these groups - for which there is plenty of literature - perhaps also worthy of discussion here.

I assume that the differences in the distributions of student ethnicity between the participating schools and the national average are because participating schools were also more likely to be urban rather than rural schools, which is also unremarked upon.

Author response: We agree that this might be one explanation, but it is worth noting that just because the non-participating schools have a higher proportion of ethnic minority students, that is not necessarily why the schools said no (particularly as ethnic minority teachers are typically under-represented in leadership roles). Without individual level data on participation it is difficult to draw firm conclusions on reasons for school-level response differences. We have included this limitation in the Discussion (page 17, lines 499-500), as well as consideration of factors that relate to lack of diversity in our study (page 18 lines 509-528).

Reviewer 3 comment 3: Was there any data on the ethnicity of the participating students for each school? Is it not also a limitation that ethnic minority students are grouped into a single category? Was there more detailed information on students' backgrounds? To gain the trust of underrepresented groups, should researchers make more effort to recognise specific racialised groups, as well as their needs and concerns? How does racism impact on willingness to participate in studies? How does racial abuse impact mental health? A stronger argument on the need for representative samples could have been achieved in the manuscript. Data from underrepresented groups are essential to ensure that healthcare interventions from such research benefit everyone equally.

Author response: We did not collect data on ethnicity of participating students. As a feasibility study, we aimed to achieve diversity in their sample rather than representativeness, as the intention was not to generalise but to understand feasibility issues in a small but diverse set of contexts to inform decisions going forward. Nevertheless, we acknowledge the lack of diversity of our sample in the limitations (page 18, lines 509-520).

We have also highlighted the need to work with individuals from underrepresented communities in future research: "In future, it will be important to understand more about recruitment and retention in ethnically diverse populations and develop research protocols that ensure that traditionally underrepresented groups are closely involved in the co-production of the research. This is particularly important for health-related research to ensure that research findings are relevant to marginalised groups who often have a high burden of mental health difficulties. It is also important so that policy and practice recommendations that follow from research are developed appropriately and fairly. We would argue that a co-production approach to genetic mental health research is essential, and that a priority is to find ways to develop new research of this kind that deals explicitly with potential barriers to participation with input from marginalised groups from the outset." (page 18, lines 520-528).

Reviewer 3 comment 4: As such, further discussion is required on the need for representative samples, and the challenges that researchers face in recruiting families from ethnic minority backgrounds and those with less socioeconomically advantaged backgrounds. Given that this is a common limitation across several British cohort studies, I wondered whether these findings could be compared with previous findings.

Author response: We have added to the discussion on the need for representative samples, and referred to relevant literature in this area (page 18, lines 509-528).

Reviewer 3 comment 5: Much of the issue of participation also hinges on the public trust of scientists and governments, to handle their data responsibly and use it for the right purposes. How can researchers use education to increase levels of understanding and trust? Are there examples in other countries that we can learn from?

Author response: We have briefly added to discussion of this on page 18, lines 536-541.

Reviewer 3 comment 6: Related to this, I wondered how the science workshop and various education materials were developed and designed? Were public engagement specialists consulted? Would this help? How could a larger audience be engaged with educational material?

I wondered what could be learned from TV shows such as “Educating...”, “The secret life of 4 and 5-year-olds” and the works of science communicators like Dr Adam Rutherford and Dr Hannah Fry in communicating the themes highlighted in this manuscript to the public. Given the enormous costs of mental health on society due to its effect on educational, occupational and life outcomes, perhaps TV and radio media could be produced to communicate these issues to a wider audience.

Author response: The science workshop was developed by MAGES researchers with consultation from teachers, SHRN researchers and experts in child and adolescent psychiatry. We also received feedback from a group of 10-13 year old boys (see “Development phase” pages 6-7) on whether the workshop was appropriate, understandable and enjoyable for this age group. Given budget constraints we were unable to engage science communication specialists. However, we agree that such initiatives would be extremely important to improve education and reduce stigma around mental health at a societal level. Unfortunately, undertaking something of this scale was beyond the scope of this small study.

Reviewer 3 comment 7: Previous studies like the GLAD study have employed social media campaigns and news interviews to increase awareness and recruit large numbers of individuals (>40,000), which are essential for genomic analyses. Perhaps a more thorough discussion of different approaches to overcome poor rates of participation would be beneficial for planning future initiatives.

Author response: We have added in the discussion the need to scale up to achieve sample sizes suitable for genomic analyses, and referenced the GLAD study as one such example (page 18, lines 529-535).

Reviewer 3 comment 8: Sample sizes required for genomic analyses were not really discussed. What should be the ambition of such initiatives? How many participants are required for genomic analyses? Furthermore, if sufficient data were obtained, which analyses would be carried out? How could this data improve outcomes for school children? I wondered whether any analyses of the SHRN phenotypic data had been conducted? And what did it show? And subsequently, why is collecting genomic data the logical next step? Is it economically justifiable? Perhaps specific examples of the influence of genetics on mental health and educational outcomes in school-aged cohorts could be referenced in the introduction to justify this direction of research.

Author response: We include more consideration of sample size within the limitations: “A further important point is that, as this study was focused on feasibility to inform future decisions, the overall sample size was small. Whilst the findings provide helpful insights on the acceptability and feasibility of the methods used, it is critical to remember that data from much large numbers of individuals are essential for standard genomic analyses. The current approach was both labour and cost-intensive and it may be that broader scale awareness raising and social media campaigns may be more

effective, such as those currently used to recruit participants to genetic mental health studies in adults.(29)” (page 18, lines 529-535).

Phenotypic data from SHRN has been extensively analysed in previous papers (see full list on the SHRN website: <https://www.shrn.org.uk/publications/>). MAGES was a separate project to assess feasibility of collecting genetic samples as well as consent to link to SHRN data and health records. Reviewer 3 comment 9: I felt that some of the teachers' testimonials could have been paraphrased such that they are less critical, more sensitive, and less repetitive: e.g. “It's the lost in translation thing – they didn't quite understand what. And some of them are quite, we have got a very, very weak [academically] to begin with and when the kids are very, very weak [academically], it was more lost in translation.” could be: “It [gets] lost in translation, they didn't quite understand what [the research involves]. Some of the students are ... weak[er] [academically] ... and [for them]... it was more lost in translation.”

Also: “I had one student ... who thought you were going to test her for mental health problems and was concerned that you were going to tell her there was something wrong with her.”

Author response: Thank you for highlighting this. We have edited some quotes taking on board these comments, ensuring quotes are less repetitive (page 14, lines 407-409; page 15 lines 413-415).

Reviewer 3 comment 10: Typing error on Page 21, line 24: “Parental consent is a challenge in school-based research,(19–22) and may be particularly challenging in secondary schools compared to primary schools where parental links to schools are not as strong as well as with research that covers mental health, genetics and data linkage.”

Author response: Thank you, this has now been changed (page 20, lines 586-589).

VERSION 2 – REVIEW

REVIEWER	von Schantz, Malcolm University of Surrey, Faculty of Health and Medical Science
REVIEW RETURNED	13-Nov-2021

GENERAL COMMENTS	The authors have addressed all comments satisfactorily, although they may wish to consider including their response to my first comment in my previous review in the manuscript and not just to the rebuttal letter
---

REVIEWER	Rayner, Christopher King's College London, Institute of Psychiatry, Psychology & Neuroscience
REVIEW RETURNED	15-Nov-2021

GENERAL COMMENTS	I am satisfied with the authors' response to my comments.
---

VERSION 2 – AUTHOR RESPONSE

Reviewer 2 comment 1: The authors have addressed all comments satisfactorily, although they may wish to consider including their response to my first comment in my previous review in the manuscript and not just to the rebuttal letter.

Author response: Thank you for your comment. We now include this information in the Methods section “Science workshops began with an introduction to MAGES and the team followed by an interactive lesson (see Figure 2) consisting of 1) a presentation teaching the basics of DNA, 2) a practical experiment extracting DNA from bananas, 3) an additional presentation on DNA structure, heredity, traits influenced by genes and impact of environment/experience, as well as an interactive discussion on non-visible traits that might be related to variation in base pair sequences (this was directed by student responses but often covered traits such as mental health, IQ, talents in sports and music, and personality), and 4) an activity creating origami DNA models.” (pages 7-8, lines 177-184)